# Oral toxicity of arjunolic acid on hematological, biochemical and histopathological investigations in female Sprague Dawley rats

Khurram Aamir[1], Hidayat Ullah Khan[1], Chowdhury Faiz Hossain[2], Mst. Rejina Afrin[2], Imam Shaik[3], Naguib Salleh[4], Nelli Giribabu[4] and Aditya Arya[5,6,7]

[1] School of Pharmacy, Faculty of Health and Medical Sciences, Taylor's University, Subang Jaya, Malaysia
[2] Department of Pharmacy, Faculty of Science and Engineering, East West University, Dhaka, Bangladesh
[3] Department of Pathology, School of Medicine, Faculty of Health and Medical Sciences, Taylor's University, Subang Jaya, Malaysia
[4] Department of Physiology, Faculty of Medicine, University of Malaya, Kuala Lumpur, Malaysia
[5] Department of Pharmacology and Therapeutics, School of Medicine, Faculty of Health and Medical Sciences, Taylor's University, Subang Jaya, Malaysia
[6] Department of Pharmacology and Therapeutics, Faculty of Medicine, Dentistry and Health Sciences, University of Melbourne, Melbourne, Parkville, VIC, Australia
[7] Malaysian Institute of Pharmaceuticals and Nutraceuticals (IPHARM), Bukit Gambir, Gelugor, Pulau Pinang, Malaysia

## ABSTRACT

**Background:** Arjunolic acid (AA) is a potent phytochemical with wider pharmacological activities. Despite potential medicinal properties on various in vitro and in vivo studies, there is still a dearth of scientific data related to its safety profile and toxicological parameters. The current study aimed to investigate acute toxicity of AA in normal female Sprague Dawley rats.

**Methods:** In this study, AA was administered orally at an individual dose of 300 and 2000 mg/kg body weight to group 1 and 2 respectively, while group 3 served as normal control. All the animals were observed for 2 weeks to determine any behavioral and physical changes. On day 15, blood was collected for hematological and biochemical investigation, later animals from all the three groups were euthanized to harvest and store essential organs for histopathological analysis. Four different staining techniques; hematoxylin and eosin, Masson trichrome, Periodic acid Schiff and Oil O Red were used to investigate any alterations in different tissues through microscopical observation.

**Results:** The results of the study showed no morbidity and mortality at two different dosage of AA treatment. Daily food & water intake, body weight, relative organ weight, hematological and biochemical parameters were detected to be normal with no severe alteration seen through microscopical investigation in the structure of harvested tissues. Our findings support the safety profile of AA, which was well tolerated at higher dose. Thus, an in-detail study on the subacute disease model is warranted.

Corresponding author
Aditya Arya,
aditya.arya@taylors.edu.my

**How to cite this article** Aamir K, Khan HU, Hossain CF, Afrin MR, Shaik I, Salleh N, Giribabu N, Arya A. 2019. Oral toxicity of arjunolic acid on hematological, biochemical and histopathological investigations in female Sprague Dawley rats. **PeerJ 7:e8045**
**DOI 10.7717/peerj.8045**

## INTRODUCTION

Nature has provided many therapeutic agents which are hidden in different forms in natural habitats. Essential compounds from marine, animal and plant sources play pivotal role in the prevention and management of various diseases. The initial concept of utilizing food as a source of medicine seems to be of great values after intensive research in the field of ethnopharmacology. Management plans and strategies using phytochemicals and medicinal plants are traditional way of treating various disorders among local communities of the past. Interestingly, usage of herbs and phytochemicals are in practice under Western herbal medicine, in Indian systems of medicine such as Ayurvedic, Unani, Siddha, and Homeopathic medicine, and in traditional Chinese medicine (*Pariyani et al., 2015*).

In the recent years more focus is appeared on the treatment of numerous clinical conditions with nutraceuticals and dietary supplements including herbal medicines. Investigation of various plant extracts and their isolated phytocompounds and botanicals are of great attention among researchers. The use of natural products has developed more interest and confidence due to their safety and efficacy among people of all age groups (*Amos et al., 2015*). However, there is a lack of scientific data on the evidence based platform as well as toxicological investigations of these natural medicines (*Yang et al., 2019*). The toxicological information and safety profile of new compounds is of prime value, enabling us to choose an appropriate dosage in animal studies at preclinical level. These findings may be applicable further on humans at later stages of clinical trials (*Thelingwani & Masimirembwa, 2014*).

Bark of *Terminalia arjuna* tree from the family of Combretaceae is regarded as a well-known herb from centuries in Ayurvedic system of medicine. The whole plant is a rich source of various active ingredients which are classified as saponins, ellagic acid, tannins, triterpenoid saponin, oligomeric proanthocyanidins, flavonoids, gallic acid and phytosterols (*Ghosh et al., 2010b*). The triterpene saponins which comprises of arjunolic acid, arjungenin, arjunic acid, and flavonoids including arjunolone, arjunone and luteolin are of great medicinal value (*Facundo et al., 2005*; *Ghosh et al., 2010b*).

Arjunolic acid (AA: 2.3,23-trihydroxyolean-12-oic acid), found in nature as chiral triterpenoid saponin which is isolated from the bark of *T. arjuna*. This compound possess variety of biological activities like, antiasthmatic (*Kalola & Rajani, 2006*), antitumor (*Wille et al., 2001*), wound healing (*Chaudhari & Mengi, 2006*), antifungal (*Masoko et al., 2008*), antibacterial (*Djoukeng et al., 2005*), and inhibition of insects growth (*Bhakuni et al., 2002*). Despite variety of biological activities, AA is well-known for its cardioprotective role and proved to be beneficial against platelet aggregation and in lowering of blood pressure, lipid level, myocardial necrosis and coagulation and heart rate (*Ghosh & Sil, 2013*). Its beneficial effects might be due to the potent antioxidant activity, which is demonstrated by its free radical scavenging activity. This compound has shown to be effective in

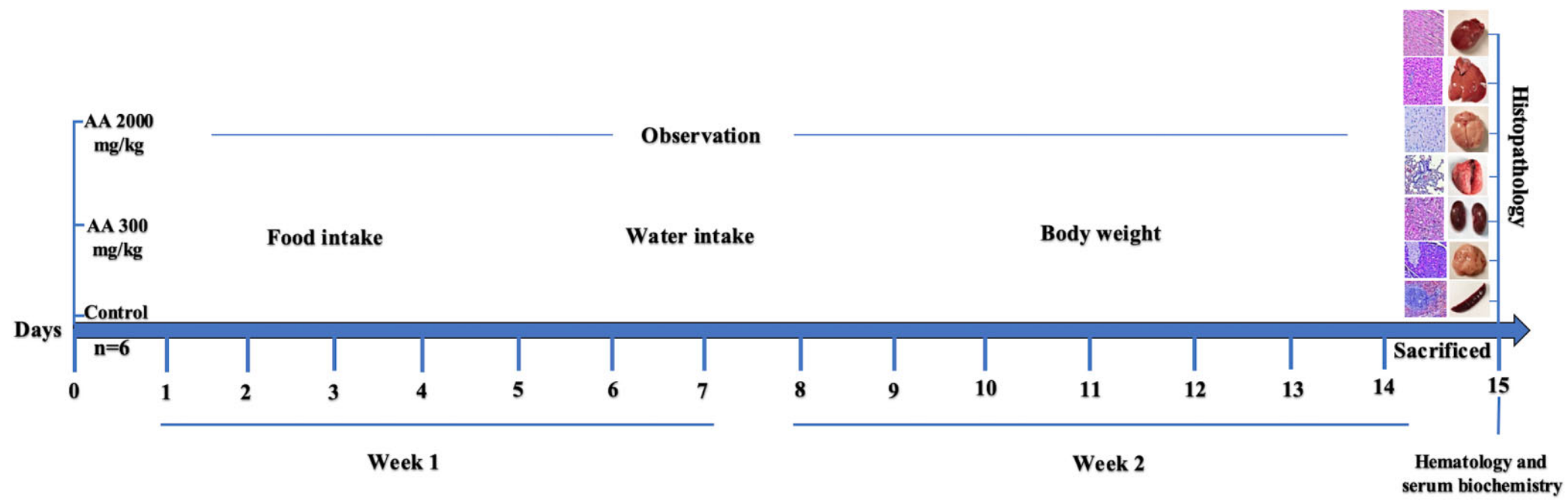

**Figure 1 Study design.**

eliminating radicals produced due to nitric oxide, superoxide and hydroxyl at the cellular level (*Ghosh & Sil, 2013*; *Manna et al., 2007*). Moreover, it possess protective effects toward cells and tissues against toxicity induced by drugs or heavy metals (*Ghosh et al., 2010a*; *Manna et al., 2007*). Various biological activities have shown its prominent use in different diseases model but its safety profile on living system is still not available. Therefore, the current study is aimed to investigate oral acute toxicity of AA in female Sprague Dawley (SD) rats to elucidate its therapeutic dose and safety in animals. The graphical representation of the study design is represented in Fig. 1.

# MATERIALS AND METHODS

## General experimental procedure

Organic solvents for extraction and chromatographic separation were obtained from Active Fine Chemicals, Bangladesh. Melting points were determined on a digital melting point Apparatus of Cole-Parmer Ltd., UK (model SMP10). Thin layer chromatography (TLC) was run on Merck pre-coated TLC plates with $Si_{60}F_{254}$. Plates were visualized by spraying with Lieberman–Burchard reagent followed by heating. Vacuum Liquid chromatography was done using Silica gel 60 (0.040–0.005 mm), Merck, Germany. Open column chromatography was performed using Silica gel 60 (0.063–0.020 mm), Merck, Germany. Spectral data were obtained as follows: Infrared (IR) spectrum with a Shimadzu IR Prestige-2 FT-IR spectrophotometer, ultraviolet spectrum with a Shimadzu UV spectrophotometer (UV-1800), nuclear magnetic resonance (NMR) spectra with an ultra-shield Bruker Avance 400 MHZ in $CD_3OD$. The NMR spectra were recorded running gradients and using residual solvent peak (at 3.33 in $^1$H-NMR and middle peak of septate at 49.0 in $^{13}$C-NMR) as internal reference.

## Plant collection

Dried barks of *T. arjuna* were collected from Bogura, a northern district of Bangladesh (Latitude: 24°51′3.53″N and Longitude: 89°22′15.89″E) by Green Herbal Supply, Shapla Chattar, Motejheel, Dhaka 1000, Bangladesh. The plant materials were authenticated by a botanist of the Department of Pharmacy at East West University, Bangladesh. A voucher specimen of the dried bark of plant is on deposit at the East West University herbarium (voucher # EWUH-PHRM-180001).

## Isolation of Arjunolic acid

Dried barks of *T. arjuna* were pulverized with a commercial grinding mill, and the coarse powder were kept in air-tight containers. A pilot scale extraction and an isolation unit were set up at Department of Pharmacy, East West University. Few batches of coarse powder of the bark (3.0 kg) were exhaustively extracted with EtOAc by Soxhlet apparatus using 10 thimbles. Each thimble contained 300 g of powder and was run 45 cycles (one cycle required 15 min). The extracting solvent was filtered, and the filtrate was concentrated under reduced pressure by a rotary evaporator (48 °C) to obtain crude EtOAc extract (53.5 g, yield: 2.7% w/w from dried sample). This crude extract was chromatographed on silica gel by an open column (4.0 × 100 cm) with step-gradient of petroleum ether (b.p. 60–80 °C) and EtOAc solvent system which yielded 86 fractions. Based on the analytical TLC (solvent system: EtOAc-MeOH-AcOH 9.0:1.0:0.1) of the fractions, these sub-fractions were then pooled into 16 fractions. The fr.13 (3.9 g), rich in triterpenoids, were further chromatographed on silica open column (4.0 × 60 cm) with step-gradient of $CHCl_3$-MeOH to get 11 fractions. Further crystallization of fr.7 to fr.9 with EtOAc yielded arjunolic acid as shown in Fig. 2.

## Experimental animals

Acute oral toxicity test was conducted following the Organization for Economic Cooperation and Development (OECD) guideline, that is, OECD Guideline 423 for the acute oral toxicity (*OECD, 2001*). Eighteen female SD rats of 8–10 weeks with an average weight of 270 ± 15 g were obtained from the Animal Experimental Unit (AEU), Faculty of Medicine, University of Malaya, Kuala Lumpur, Malaysia. All the animals were acclimatized for 7 days with six rats in a group were sheltered in three different groups before starting acute toxicity experiments. The temperature of animal house was maintained at 24 °C, with a 12 h light/dark cycle. All experimental procedure of animal handling was approved by the Institutional Animal Care and Use Committee (IACUC) with Ethics No. 2018-210605/TAY/R/AA (2018294), Faculty of Medicine, University of Malaya. Body weight of all the animals was recorded on individual basis before administration of the test compound (AA). The volume of the test dose was calculated based on the body weight of the individual rat. All rats used were nulliparous and non-gravid, they were fed with standard pellet diet and water ad libitum.

## Acute oral toxicity test

Arjunolic acid (AA) was administered by oral gavage to overnight fasted female rats by suspending it in 0.5% carboxymethyl cellulose (CMC) in a total volume of 10 ml/kg body weight. The first test dose of 300 mg/kg body weight of AA was orally administered to the group 1. All the rats were carefully monitored for any changes in general behavioural, signs of toxicity and subsequent mortality after treatment with 1st dose for 4 h, followed by an observation period of 48 h. Group 2 animals were administered with high dose of AA at 2,000 mg/kg body weight, after 48 h. At the same time, the 3rd group (standard control) was treated with vehicle (0.5% CMC) for comparative analysis based on the OECD guideline (*OECD, 2001*). All rats were critically monitored throughout the study

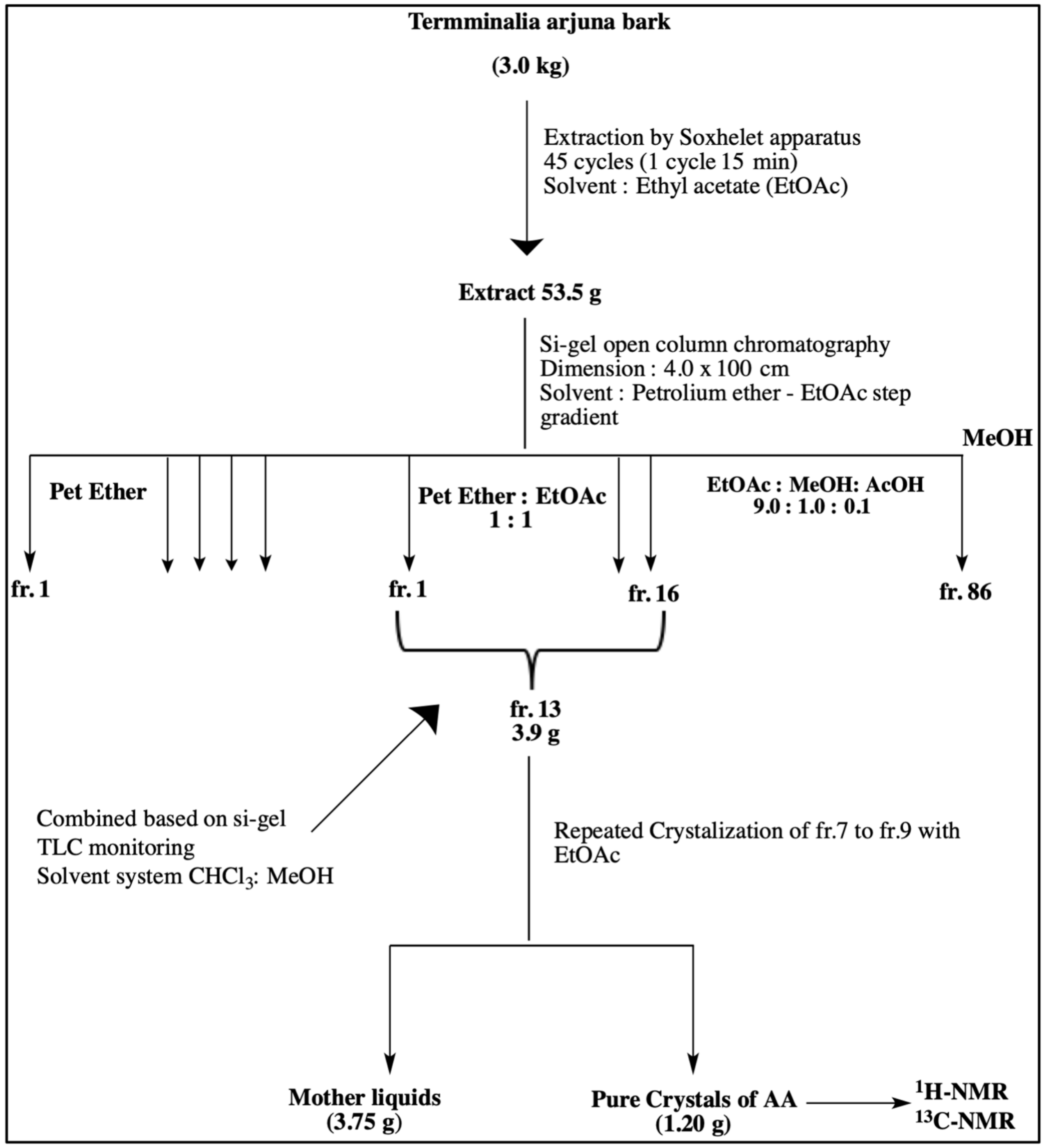

**Figure 2 Schematic diagram representing isolation of Arjunolic acid (AA).** Schematic diagram representing isolation of Arjunolic acid (AA). Ethyl acetate (EtOAc), Petroleum ether (Pet Ether), Methyl alcohol (MeOH), Acetic acid (AcOH), Thin layer chromatography (TLC), Fraction (fr), Nuclear magnetic resonance (NMR).

period and were carefully monitored for the first 30 min followed by 4 h after AA and vehicle administration and finally once in 24 h for 2 weeks. Animals were on observation for any changes in the fur, skin, mucous membranes, eyes and monitored for any behavioral changes on regular basis to maintain proper record. Moreover, special attention was given to check any signs of tremors, convulsions, salivation, lethargy, diarrhoea, coma, sleep and mortality.

The percentage variation in body weight of rats was calculated as per the body weight prior to the administration of the test compound and then weekly for

2 weeks. The following equation was used to compute the percentage change in body weight.

$$\%\text{Body weight change} = \frac{\text{Body weight at the end of each week} - \text{Initial body weight}}{\text{Initial body weight}} \times 100$$

## Hematology and serum biochemistry

At the end of study period (15th day), rats from all the three groups were euthanized by using appropriate dose of ketamine (80 mg/kg) and xylazine (7 mg/kg) following the established protocol by AEU, University of Malaya. Blood (5 ml) was collected by direct cardiac puncture and preserved in EDTA coated and plain vacutainers for hematological analysis; red blood cells (RBCs), white blood cells (WBCs), monocytes, neutrophils, lymphocytes, basophils and eosinophils. Moreover, for biochemical analysis, plain vacutainers were kept at room temperature for 15–20 min to let blood coagulate. Later blood samples were centrifuged at 5,000 rpm for 20 min at 4 °C to obtain serum for renal function tests including calcium, uric acid, sodium, potassium, chloride, urea, creatinine, inorganic phosphate, and lipid profiles; HDL, LDL cholesterol, total cholesterol and triglycerides, as well as hepatic function test mainly total protein, albumin, globulin, total bilirubin, aspartate aminotransferase (AST), alanine aminotransferase (ALT).

## Histopathological observation

Veterinarian from AEU, University of Malaya performed necropsies on all the animals complying established protocol. Subsequently, vital organs; heart, liver, lungs, kidneys, brain along with spleen, uterus, adrenal glands, stomach, ovaries, cervices, vagina, urinary bladder and eyes were harvested by making fine incision on the midline. Blood was wiped from the harvested tissues through the blotting paper and calibrated weighing balance was used to measure relative organ weight (ROW). The ROW was computed with respect to body weight at the time of sacrifice with the help of the following equation:

$$\text{ROW} = \frac{\text{Absolute weight of organ}}{\text{Body weight at the time of sacrifice}} \times 100$$

Later, heart, liver, lungs, kidney, brain, spleen and pancreas were carefully fixed in 10% buffered solution of formalin and further processed by dehydrating in graded alcohol and xylene. The processed tissues were embedded in paraffin wax for histopathological examination. Importantly, for Oil O Red staining, sections of frozen tissues were used. After embedding process, sections of paraffin were cut into fine sections of five µm, shifted on glass slides and stained with hematoxylin and eosin (H&E). In order to investigate fibrotic lesions in tissues, sections were stained with Massion trichrome (Abcam 150686), while accumulation of mucopolysaccharides and fats were examined by staining with Periodic acid Schiff (PAS) (Abcam 150680) and Oil O red (Abcam 150681) respectively as per the manufacturer protocol. Tissue sections were further observed under the fluorescence compound microscope (Eclipse Ni-U, Nikon Corporation, Tokyo, Japan) for any kind of microscopic variations in the harvested tissues.

### Statistical analysis

Results were expressed as the mean ± standard deviation (SD). Normality of data was analyzed by Kolmogorov–Smirnov and Shapiro–Wilk test. The statistical variations between groups were analyzed by one-way analysis of variance (ANOVA) using SPSS package version 22 (IBM, SPSS Statistics, Inc., Chicago, IL, USA). Differences were considered significant at ($p < 0.05$).

## RESULTS

### Arjunolic acid

Crystallization of fr. 7 to fr. 9 with EtOAc yielded AA 1.20 g as shown in Fig. 2. The obtained crystals were white amorphous powder; m.p: >300 °C (decom.); UV $\lambda_{(MeOH)}$ nm: 204 (end absorption); IR $\nu_{(KBr)}$ $cm^{-1}$: 1,047.4, 1,197.9, 1,386.9, 1,461.1, 1,637.6, 1,694.5, 2,645.5, 2,884.7, 2,942.5, 3,416.1; $^{1}$H-NMR ($CD_3OD$):: 0.72 (3H, s, $H_3$-24), 0.83 (3H, s, $H_3$-25), 0.87–0.92 (1H, m), 0.96 (3H, s, $H_3$-26) 0.97 (3H, s, $H_3$-30), 1.02 (3H, s, $H_3$-29), 1.03 (3H, s, $H_3$-27), 1.31–1.35 (4H, m), 1.45–1.59 (3H, m), 1.67–1.81 (6H, m), 1.91–1.22 (1H, m), 2.00 (1H, d, $J$ = 8.8 Hz), 2.33 (1H, dd, $J$ = 4, 4 Hz) 3.28–3.43 (7H, m) 2.92 (1H, d, $J$ = 9.6) 3.08 (1H, br s) 3.61 (1H, m, $H_b$-11) 3.69 (1H, dd, $J$ = 4.0, 8 Hz, H-2), 3.84 (1H, d, $J$ = 8 Hz, H-3), 5.40 (1H, d, $J$ = 4 Hz, H-12); $^{13}$C-NMR ($CD_3OD$): 48.05 (C-1), 69.51 (C-2) 78.33 (C-3), 43.07 (C-4), 49.06 (C-5) 19.70 (C-6), 33.28 (C-7), 40.50 (C-8), 49.21 (C-9) 39.38 (C-10), 24.90 (C-11), 124.81 (C-12), 144.43 (C-13), 42.67 (C-14), 29.43 (C-15), 28.44 (C-16), 47.12 (C-17), 45.06 (C-18), 48.02 (C-19), 31.52 (C-20), 33.81 (C-21), 29.51 (C-22), 62.40 (C-23), 17.01 (C-24) , 17.60 (C-25), 17.37 (C-26), 25.17 (C-27), 178.58 (C-28). 26.35 (C-29), 24.98 (C-30).

### Analysis of general sign and behavior

Oral administration of AA at 300 and 2,000 mg/kg body weight showed no signs of mortality. During the whole study period no signs of toxicity was observed. No alterations in eyes, skin, fur, mucous membrane and behavioral patterns were noted. Moreover, there were no signs of diarrhoea, salivation, coma, sleep, lethargy, and tremors as shown in Table 1. Based on the general observation from the study, AA did not show any lethal effect on animals at tested dose levels.

### Effect of AA on body weight, relative organ weight, food and water consumption

The average body weight of normal animals was observed as 263.33 ± 12.11 gm, whereas AA (300 and 2,000 mg/kg) showed 263.16 ± 2.48 and 251.50 ± 8.91 gm respectively. Moreover, increase in the body weight was observed in the treatment groups, which shows non-significant difference when compared to control group ($p > 0.05$), as stated in Table 2. Furthermore, all the harvested organs were recorded normal weight and regular morphology without any macroscopic lesions as shown in Figs. 3 and 4, while ROW between treated and control groups exhibited non-significant variations ($p > 0.05$) presented in Table 3.

**Table 1** General observation and behavioral analysis.

| Observation | Control | | AA (300 mg/kg) | | AA (2,000 mg/kg) | |
|---|---|---|---|---|---|---|
| | 4 h | 24 h | 4 h | 24 h | 4 h | 24 h |
| Eyes | NC** | NC | NC | NC | NC | NC |
| Skin and fur | NC | NC | NC | NC | NC | NC |
| Lethargy | NO | NO | NO | NO | NO | NO |
| Sleep | Normal | Normal | Normal | Normal | Normal | Normal |
| Diarrhea | NO* | NO | NO | NO | NO | NO |
| Coma | NO | NO | NO | NO | NO | NO |
| Tremors | NO | NO | NO | NO | NO | NO |
| Mucous membrane | NC | NC | NC | NC | NC | NC |
| Behavioral patterns | Normal | Normal | Normal | Normal | Normal | Normal |
| Salivation | Normal | Normal | Normal | Normal | Normal | Normal |

**Notes:**
* Not observed
** No change

**Table 2** Percentage gain in body weight (g) of rats at each week.

| Week | Control | AA (300 mg/kg) | AA (2,000 mg/kg) | *p* value |
|---|---|---|---|---|
| Week 1 (%) | 5.66 ± 1.65 | 5.90 ± 2.78 | 9.29 ± 4.58 | 0.13 |
| Week 2 (%) | 12.74 ± 1.64 | 11.10 ± 2.73 | 13.17 ± 6.77 | 0.688 |

**Note:**
Values expressed as mean ± standard deviation.

Daily food and water intake of the treatment groups revealed non-significant difference ($p > 0.05$) when compared with the normal animals (Table 4).

## Effect of AA on hematological and biochemical profile

The hematological parameters are summarized in Table 5. Treatment groups showed non-significant changes ($p > 0.05$) in the hematological parameters; RBCs, hemoglobin, WBCs, packed cell volume (PCV), mean corpuscular volume (MCV), mean corpuscular hemoglobin (MCH) and mean corpuscular hemoglobin concentration (MCHC), when compared with the normal control.

Similarly, the biochemical parameters; urea, uric acid, creatinine, low density lipoproteins (LDL), high density lipoproteins (HDL), triglycerides (TG), total cholesterol (TC), alanine amino transferase (ALT/SGPT), aspartate amino transferase (AST/SGOT), alkaline phosphatase (AP), total proteins, albumin and globulin levels of the treatment groups were not affected and observed to be normal when compared with the control group ($p > 0.05$), shown in Table 6.

## Histopathological findings

Histopathological examination was carried out to investigate any structural alterations in the harvested tissues. Four different staining; H&E (Figs. 5 and 6), Masson trichrome (MT) (Figs. 7 and 8), PAS (Figs. 9 and 10) and Oil O Red (Figs. 11 and 12) were used for the microscopic investigation of the harvested tissues; heart, liver, kidneys, lungs, brain, spleen

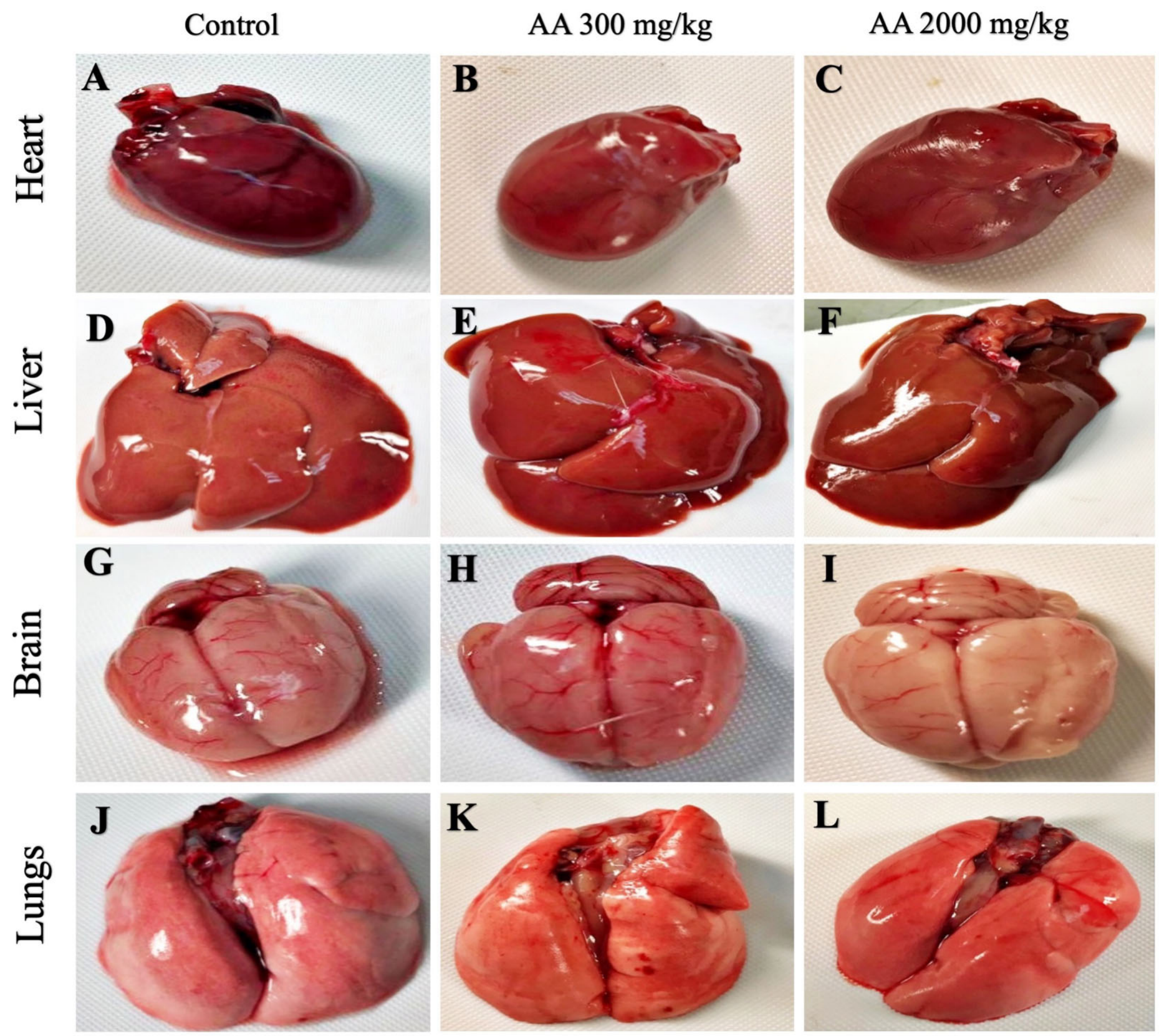

**Figure 3 Macroscopic photographs of harvested organs.** Macroscopic photographs showing normal morphology of heart (A–C), liver (D–F), brain (G–I) and lungs (J–L) from female SD rats.

and pancreas. The microscopical findings, did not express any gross pathological and microscopical changes in the treatment groups, when compared to the control animals. Moreover, during gross examination all the organs retained their normal textures without any abnormal changes in the color and appearance. Importantly, no structural alterations were detected in different tissues using specific resolution under the microscope.

## DISCUSSION

Since time immemorial plant-derived botanicals have important role in the management of various diseases. Natural products and phytochemicals have potential to develop as therapeutics or might have essential role to be used as a precursors for the synthesis of potent drug candidates (*Ifeoma & Oluwakanyinsola, 2013*). However, at the same time investigation of potential toxicity of plant derived compounds is crucial prior to their usage in various disease models.

In the present study, arjunolic acid (AA) is isolated from the bark of *T. arjuna*, which have a $R_f$ value of 0.62 with EtOAc-MeOH-AcOH 9.0:1.0:0.1 solvents system on silica gel

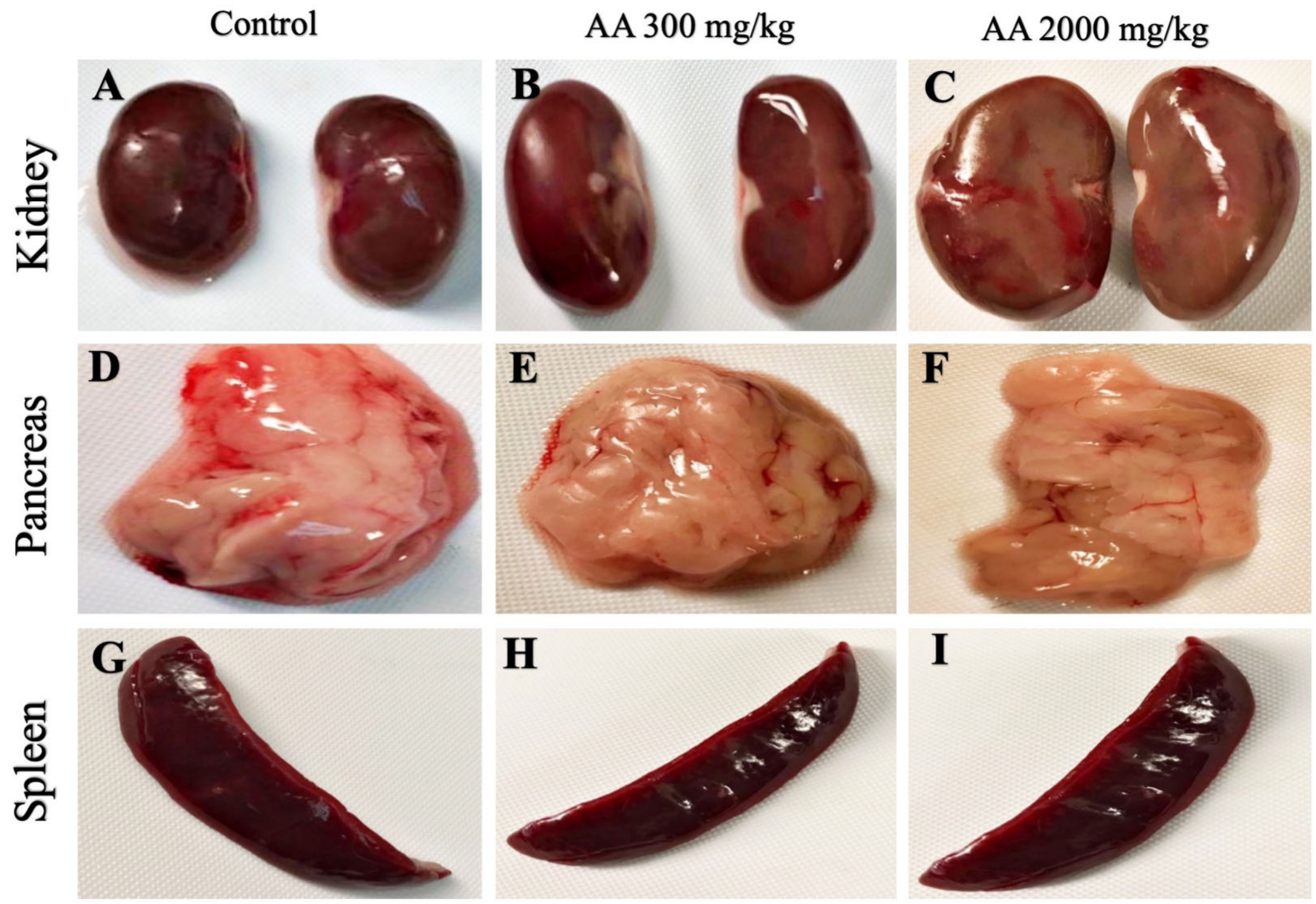

**Figure 4 Macroscopic photographs of harvested organs.** Macroscopic photographs showing normal morphology of kidney (A–C), pancreas (D–F) and spleen (G–I) from female SD rats.

**Table 3 Relative organ weight (g) of rats at time of sacrifice.**

| Organ | Control | AA (300 mg/kg) | AA (2,000 mg/kg) | *p* value |
|---|---|---|---|---|
| Brain | 0.632 ± 0.035 | 0.750 ± 0.121 | 0.694 ± 0.083 | 0.098 |
| Heart | 0.267 ± 0.001 | 0.269 ± 0.002 | 0.268 ± 0.003 | 0.359 |
| Stomach | 0.779 ± 0.065 | 0.815 ± 0.200 | 0.861 ± 0.201 | 0.704 |
| Lungs | 0.433 ± 0.002 | 0.432 ± 0.003 | 0.433 ± 0.002 | 0.784 |
| Liver | 2.947 ± 0.093 | 3.176 ± 0.363 | 3.104 ± 0.217 | 0.299 |
| Pancreas | 0.447 ± 0.003 | 0.446 ± 0.003 | 0.445 ± 0.004 | 0.703 |
| Spleen | 0.217 ± 0.021 | 0.255 ± 0.055 | 0.233 ± 0.030 | 0.271 |
| Left Kidney | 0.262 ± 0.001 | 0.260 ± 0.002 | 0.262 ± 0.001 | 0.132 |
| Right Kidney | 0.262 ± 0.002 | 0.261 ± 0.003 | 0.263 ± 0.003 | 0.489 |
| Left Ad gland | 0.021 ± 0.004 | 0.021 ± 0.003 | 0.023 ± 0.003 | 0.579 |
| Right Ad gland | 0.020 ± 0.003 | 0.019 ± 0.003 | 0.024 ± 0.005 | 0.144 |
| Urinary bladder | 0.033 ± 0.002 | 0.032 ± 0.002 | 0.033 ± 0.003 | 0.880 |
| Uterus | 0.427 ± 0.051 | 0.526 ± 0.178 | 0.471 ± 0.057 | 0.342 |
| Cer/Vag | 0.066 ± 0.002 | 0.065 ± 0.002 | 0.067 ± 0.003 | 0.504 |
| Left Ovary | 0.036 ± 0.001 | 0.036 ± 0.003 | 0.036 ± 0.003 | 0.936 |
| Right Ovary | 0.037 ± 0.001 | 0.036 ± 0.001 | 0.035 ± 0.002 | 0.186 |
| Left Eye | 0.041 ± 0.003 | 0.044 ± 0.003 | 0.045 ± 0.005 | 0.238 |
| Right Eye | 0.042 ± 0.003 | 0.042 ± 0.003 | 0.043 ± 0.006 | 0.833 |

**Note:**
Values expressed as mean ± standard deviation.

**Table 4** Food consumption (g) and water intake (mL) by rats.

| Week | Control | AA (300 mg/kg) | AA (2,000 mg/kg) | *p* value |
|---|---|---|---|---|
| Food intake | | | | |
| Week 1 | 140.14 ± 2.193 | 140.14 ± 1.069 | 139.57 ± 1.272 | 0.884 |
| Week 2 | 140.00 ± 1.290 | 139.85 ± 1.345 | 139.57 ± 2.225 | |
| Water intake | | | | |
| Week 1 | 213.00 ± 18.248 | 185.00 ± 25.658 | 193.85 ± 23.891 | 0.108 |
| Week 2 | 180.00 ± 36.968 | 162.85 ± 28.263 | 198.57 ± 48.795 | |

**Note:**
Values expressed as mean ± standard deviation.

**Table 5** Hematological profile of rats.

| Hematological parameters | Control | AA (300 mg/kg) | AA (2,000 mg/kg) | *p* value |
|---|---|---|---|---|
| RBCs ($10^{12}$/L) | 7.40 ± 0.275 | 7.30 ± 0.894 | 7.58 ± 0.407 | 0.706 |
| Hb (g/dL) | 12.73 ± 0.937 | 13.21 ± 1.738 | 13.68 ± 0.495 | 0.398 |
| WBCs ($10^{9}$/L) | 3.45 ± 0.361 | 3.03 ± 0.382 | 3.10 ± 0.352 | 0.140 |
| Platelets ($10^{9}$/L) | 1,001.66 ± 4.082 | 1,003.33 ± 5.163 | 1,003.33 ± 5.163 | 0.791 |
| Neutrophils (%) | 17.16 ± 1.471 | 23.00 ± 8.876 | 26.00 ± 4.732 | 0.056 |
| Lymphocytes (%) | 76.33 ± 7.339 | 68.50 ± 9.974 | 67.16 ± 5.913 | 0.130 |
| Monocytes (%) | 3.83 ± 0.752 | 4.66 ± 2.804 | 4.66 ± 0.816 | 0.641 |
| Eosinophils (%) | 3.33 ± 0.816 | 4.00 ± 1.264 | 2.66 ± 1.211 | 0.152 |
| PCV (HCT%) | 34.00 ± 0.894 | 34.06 ± 1.169 | 34.00 ± 1.264 | 0.957 |
| MCV (fL) | 58.00 ± 2.607 | 59.00 ± 1.549 | 60.66 ± 3.614 | 0.262 |
| MCH (pg) | 17.88 ± 0.228 | 17.83 ± 0.408 | 17.50 ± 0.547 | 0.288 |
| MCHC (g/dL) | 30.16 ± 0.408 | 30.33 ± 0.516 | 30.16 ± 0.408 | 0.761 |

**Note:**
Values expressed as mean ± standard deviation.

**Table 6** Biochemical profile of rats.

| Biochemical parameters | Control | AA (300 mg/kg) | AA (2,000 mg/kg) | *p* value |
|---|---|---|---|---|
| Urea (mmol/L) | 11.31 ± 2.668 | 11.55 ± 1.953 | 9.76 ± 0.725 | 0.260 |
| Creatinine (Umol/L) | 46.50 ± 6.284 | 47.16 ± 10.225 | 48.83 ± 1.329 | 0.838 |
| Uric acid (mmol/L) | 0.17 ± 0.005 | 0.17 ± 0.008 | 0.16 ± 0.008 | 0.672 |
| LDL (mmol/L) | 1.50 ± 0.346 | 1.90 ± 0.525 | 1.88 ± 0.172 | 0.149 |
| HDL (mmol/L) | 1.31 ± 0.040 | 1.35 ± 0.054 | 1.30 ± 0.063 | 0.290 |
| TG (Umol/L) | 0.35 ± 0.054 | 0.36 ± 0.051 | 0.38 ± 0.040 | 0.521 |
| TC (mmol/L) | 2.88 ± 0.801 | 2.71 ± 0.519 | 2.55 ± 0.242 | 0.608 |
| Albumin (g/L) | 35.16 ± 0.752 | 35.16 ± 0.752 | 34.83 ± 0.753 | 0.682 |
| Globulin (g/L) | 31.83 ± 0.752 | 31.33 ± 0.816 | 31.33 ± 0.816 | 0.472 |
| Total Protein (g/L) | 69.00 ± 3.847 | 74.83 ± 7.833 | 73.50 ± 5.683 | 0.244 |
| AP (IU/L) | 40.50 ± 6.156 | 33.50 ± 9.792 | 51.00 ± 18.740 | 0.087 |
| SGOT/AST (IU/L) | 134.16 ± 9.948 | 127.66 ± 4.926 | 134.66 ± 7.393 | 0.246 |
| SGPT/ALT (IU/L) | 42.83 ± 5.115 | 45.50 ± 3.885 | 42.83 ± 5.600 | 0.568 |

**Note:**
Values expressed as mean ± standard deviation.

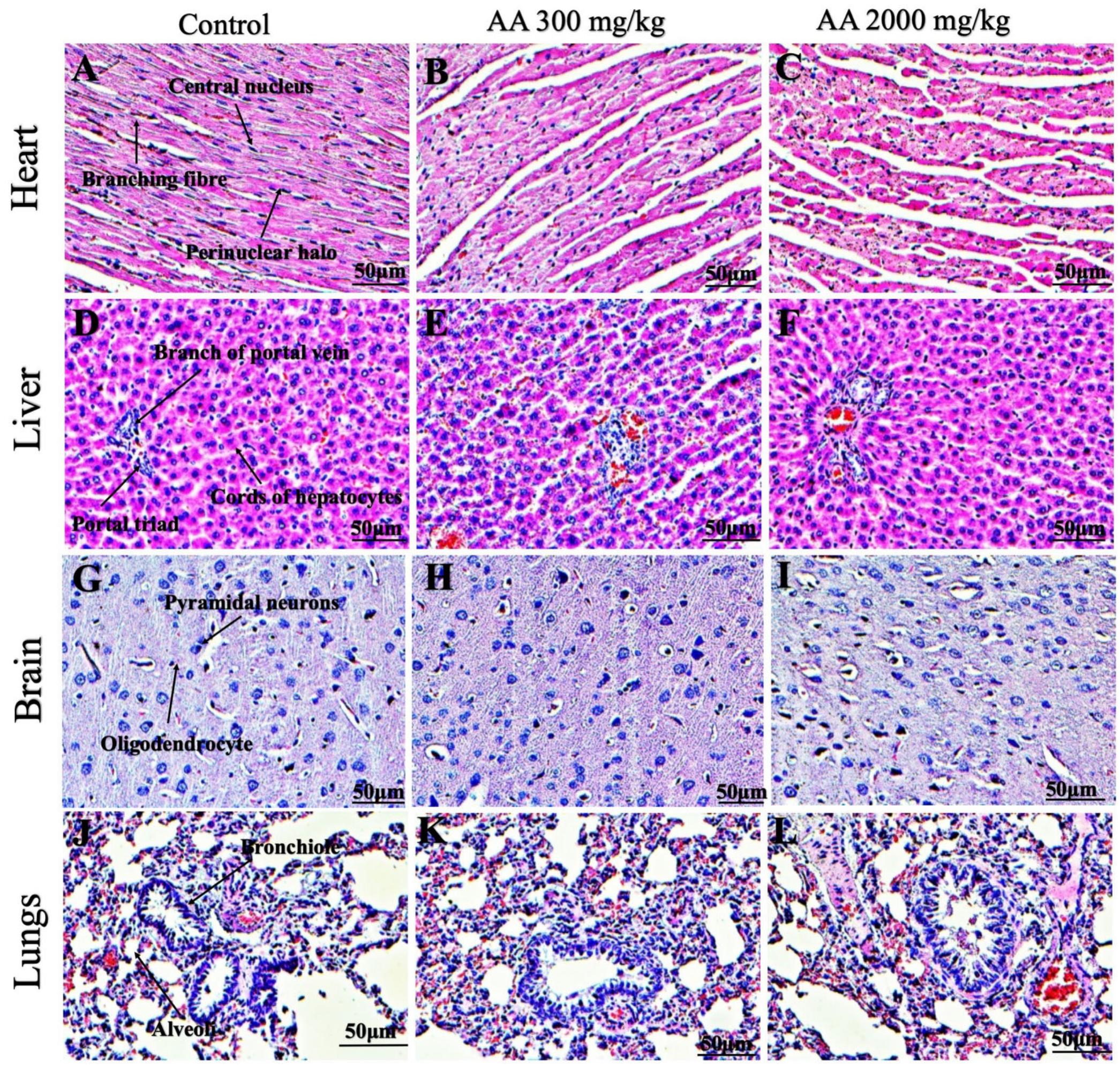

**Figure 5 Photomicrographs of vital organs displaying H&E staining of heart, liver, brain and lungs.** Photomicrographs of vital organs displaying normal architecture of heart (A–C), liver (D–F), brain (G–I) and lungs (J–L) after single oral dose of AA (H&E stain ×200).

TLC, and gave characteristic purple color of pentacyclic triterpenoid by spraying with Lieberman–Burchard reagent. It has shown broad absorption band from 2,500 $cm^{-1}$ in IR spectra which revealed, the compound is a triterpenoid acid. Strong sharp absorption band of carbonyl carbon at 1,637 $cm^{-1}$ also supported the fact. $^{1}$H-NMR spectrum of the compound showed an olefinic resonance peak at δ 5.40 (d, $J$ = 4 Hz) indicating presence of a double bond. Besides that, it showed six methyl signals at δ 0.72 (3H, s), δ 0.83 (3H, s), δ 0.96 (3H, s), δ 0.97 (3H, s), δ 1.02 (3H, s) and δ 1.03 (3H, s). $^{13}$C-NMR and DEPT (135°). Spectral analysis revealed the presence of six primary $sp^3$ carbons at δ 17.01, δ 17.37, δ 17.60, δ 24.98, δ 25.17, δ 26.35, 10 secondary $sp^3$ carbons at δ 19.70, δ 24.90, δ 28.44, δ 29.43, δ 29.51, δ 33.28, δ 33.81, δ 48.02, δ 48.05, δ 62.4, five tertiary $sp^3$ carbons at δ 45.06, δ 49.06, δ 49.21, δ 69.51, δ 78.33, and six quaternary $sp^3$ carbons at δ 31.52, δ 39.38, δ 40.5, δ 42.67, δ 43.07 and δ 47.12. Moreover, $^{13}$C-NMR spectrum also shown a tertiary $sp^2$ carbon at δ 124.81, and two quaternary $sp^2$ carbon at δ 144.43 and δ 178.58.

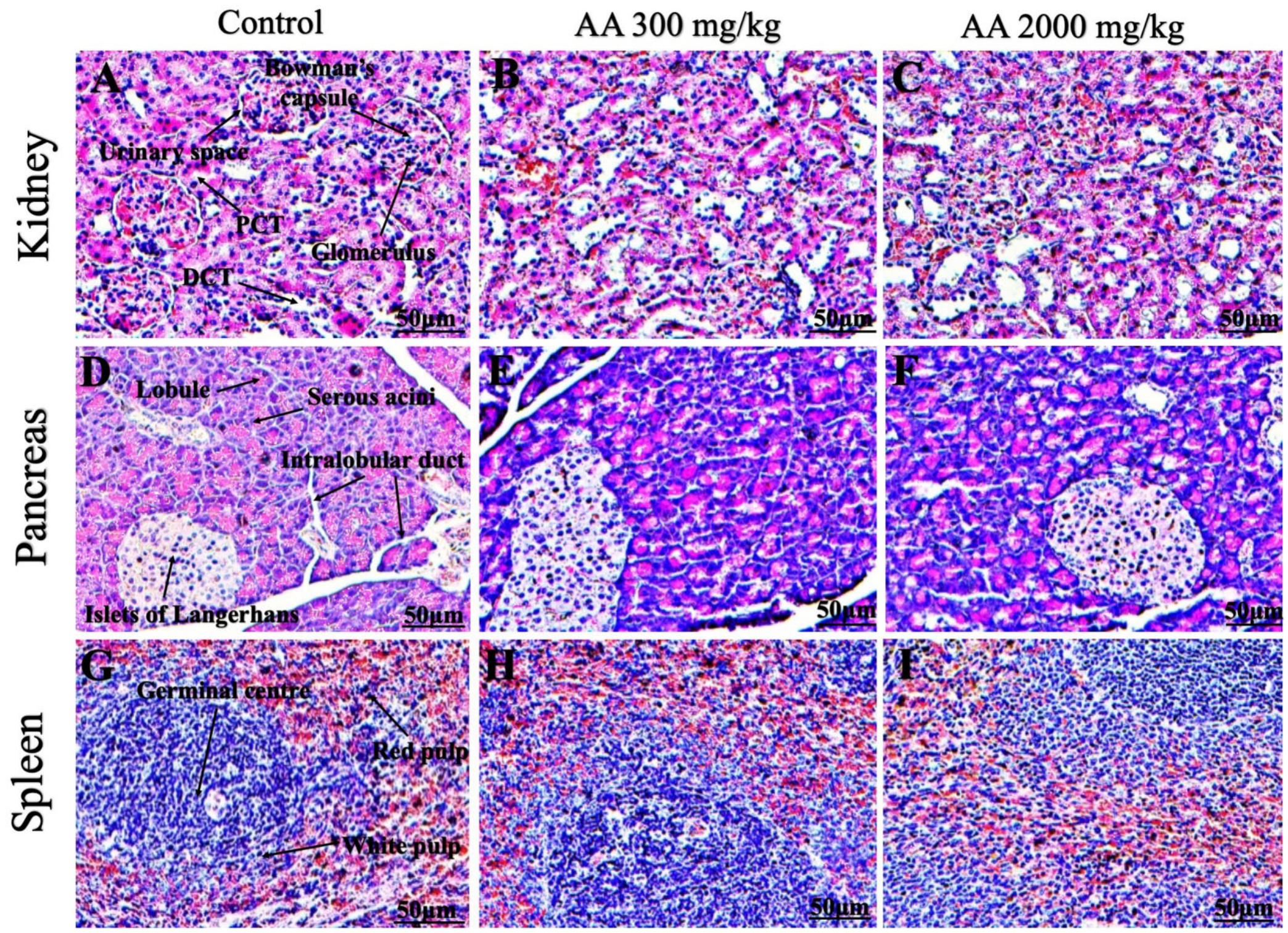

**Figure 6 Photomicrographs of vital organs representing normal architecture of kidney, pancreas and spleen.** Photomicrographs of vital organs displaying normal architecture of kidney (A–C), pancreas (D–F) and spleen (G–I) after single oral dose of AA (H&E stain ×200). Proximal convoluted tubules (PCT), Distal convoluted tubules (DCT).

Based on the spectroscopic analysis, the isolated compound was identified as arjunolic acid. Importantly, our findings are in consistent with the data published in the literature (*King, King & Ross, 1954*; *Mann et al., 2012*; *Ramesh et al., 2012*).

This study aimed to identify toxicological profile of AA which have shown to possess pleiotropic biological activities on various in vitro and in vivo study models like anti-bacterial, anti-tumor, antioxidant, antiasthmatic, anti-fungal and cardioprotective effects. However, AA has multiple pharmacological activities, the scientific data on its toxicological and safety profile in animals remains to be elucidated.

It is crucial to evaluate toxicity of any plants derived products prior to their use in laboratory animals in acute, subacute or chronic disease models (*Farsi et al., 2013*). Therefore, we investigated oral acute toxicity of AA in female SD rats with the aim to establish safe dosage for further preclinical study on subacute disease model. In this study, the selection of rat model is based on the resemblance of physiology with humans and the convenience route of administration and to integrate further pharmacokinetic parameters.

In acute toxicity study, rats were divided into three groups, one served as control (vehicle) while other two (treatment groups) received orally AA at a dose of 300 and 2,000 mg/kg based on *OECD (2001)*. Prior to AA administration, all the animals underwent fasting for approximately 12 h to avoid its interaction with food and other enzymes of digestive tract (*Kumar & Lalitha, 2013*).

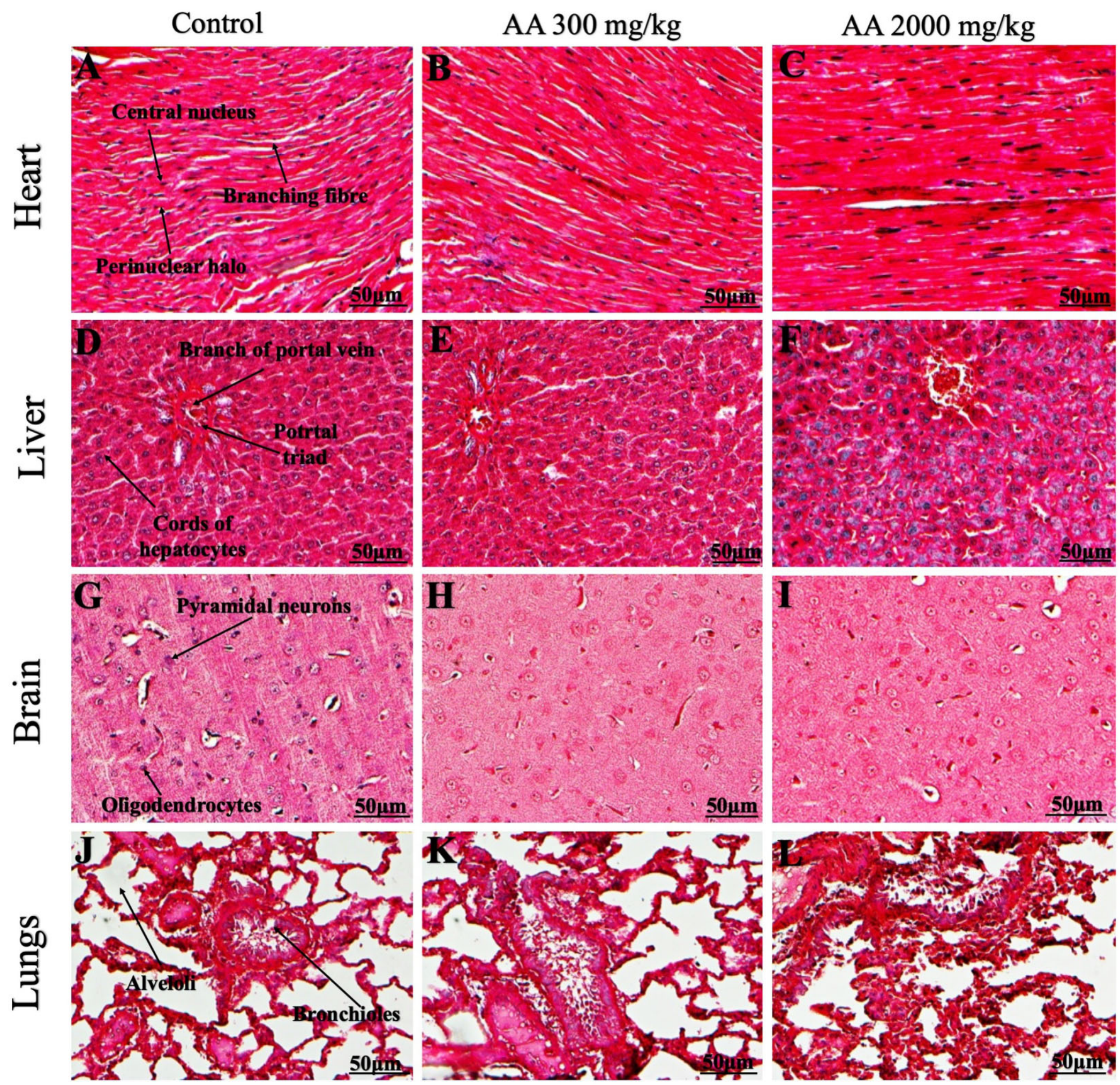

**Figure 7 Photomicrographs of vital organs presenting Masson trichrome staining of heart, liver, brain and lungs.** Photomicrographs of vital organs presenting normal architecture of heart (A–C), liver (D–F), brain (G–I) and lungs (J–L) after a single oral dose of AA (Masson trichrome stain, ×200).

No mortality was noted at the first test dose of 300 mg/kg in the treatment group which led to the administration of next high dose of 2,000 mg/kg body weight.

All the groups were monitored carefully for 2 weeks for any signs of associated toxicity and mortality. All rats in the treatment groups displayed no evident symptoms of distress. Physical observation and clinical parameters are the most imperative approach toward identification of symptoms associated with toxicity (*Jothy et al., 2011*). No major alterations in the physical characteristics of fur, skin, eyes, salivation, mucous membrane, sleep and behavioral pattern were observed. Moreover, there were no signs of diarrhoea, coma, tremors and lethargy (Table 1). In addition, both the treatment groups exhibited normal increase in the body weight, compared to the control group without any significant variation. Interestingly, our results showed non-significant changes in the ROW, as shown in Tables 2 and 3. The steady increase in the body weight and normal organ weight

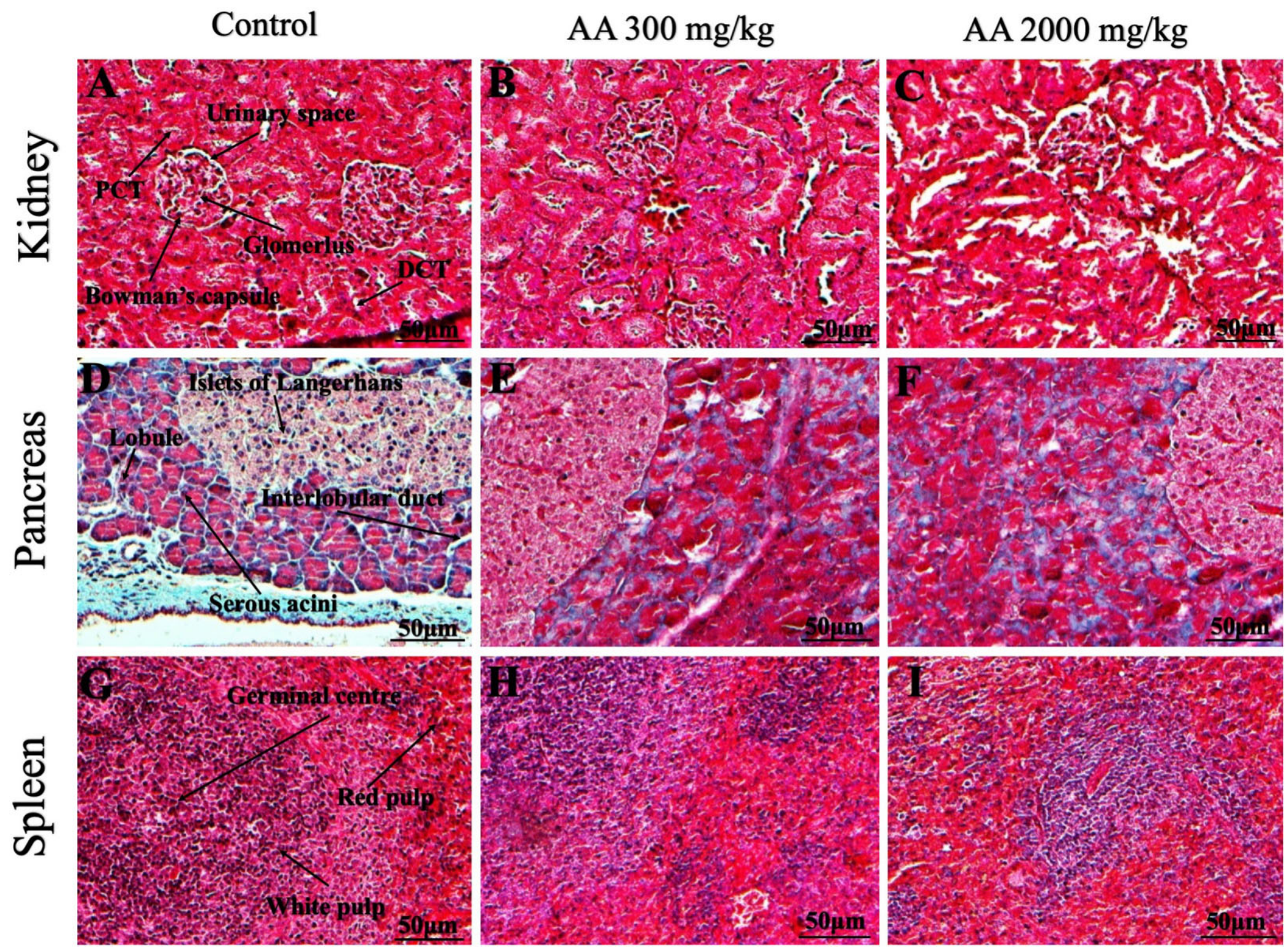

**Figure 8 Photomicrographs of vital organs presenting Masson trichrome staining of kidneys, pancreas and spleen.** Photomicrographs of vital organs presenting normal architecture of kidneys (A–C), pancreas (D–F) and spleen (G–I) after a single oral dose of AA (Masson trichrome stain, ×200). Proximal convoluted tubule (PCT), Distal convoluted tubule (DCT).

assumes AA has no devastating role in the normal growth and development process of the animals. Importantly, the organ weight is of prime importance and considered as a bench mark to body weight ratio, reflecting treatment related toxicity (*Sohail et al., 2018*).

Estimation of therapeutic window is also an important parameter that provides benefit to risk assessment of the drug/compound, animal model is one of the approach to unveil this aspect (*Al-Afifi et al., 2018*). Normal food and water intake throw light on the normal metabolic processes, as observed in the treated animals (Table 4). The results are in agreement with *Yuet Ping et al. (2013)*, who showed methanolic extract of *Euphorbia hirta* did not express any variations in the food and water intake and body weights of rats. Moreover, it is essential to evaluate food and water intake of the test compound after single oral dose to substantiate the safety, as appropriate consumptions of nutrients are valued to the health status of animals and highlight proper response to the treatment (*Sathish et al., 2012*). The results of body weight and dietary consumption shows that AA is safe even at high dose.

Result of hematological analysis revealed that AA might have normal role in the production and development process of blood cells, as the data shows non-significant alterations in the treatment vs control group (Table 5). Hematological parameters point out minor and major alterations in the functioning of different organs

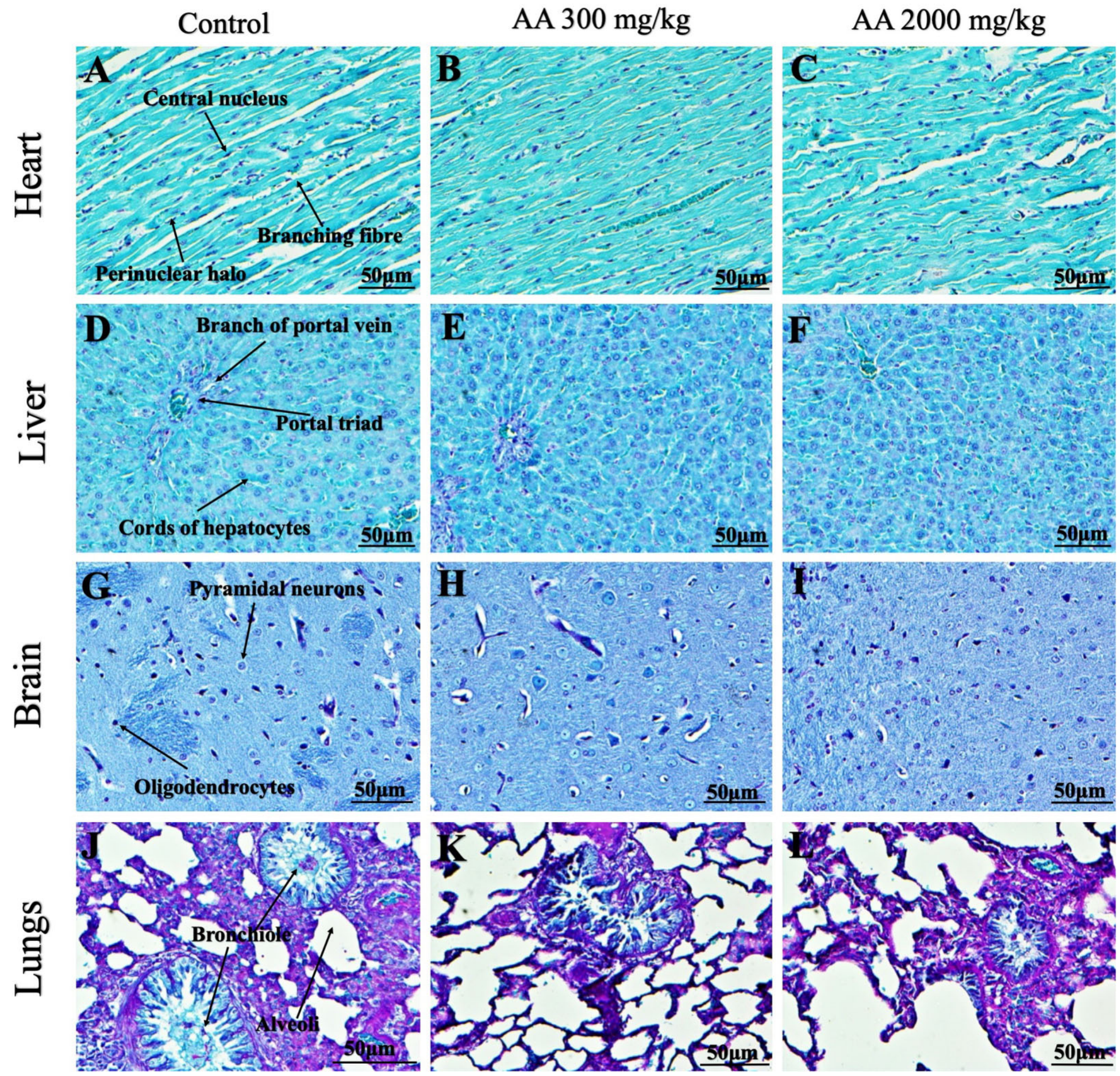

**Figure 9 Photomicrographs of vital organs demonstrating Periodic acid Schiff staining of heart, liver, brain and lungs.** Photomicrographs of vital organs demonstrating normal architecture of heart (A–C), liver (D–F), brain (G–I) and lungs (J–L) after a single oral dose of AA (Periodic acid Schiff stain, ×200).

(*Al-Afifi et al., 2018*). Moreover, our finding is supported by the previous researchers (*Petterino & Argentino-Storino, 2006*; *Said & Abiola, 2014*). The hematological parameters provide clear understanding of any signs of toxicity, as hematological system is very sensitive against the effects of any toxin entering living system and have higher diagnostic value for signs of intoxication (*Olson et al., 2000*). The foremost medium for transportation of nutrients and drugs in the living system is blood, whereas blood cells and proteins are primarily exposed to the considerable concentration of toxic compounds. Therefore, any damage to the blood components will adversely affect the smooth functioning of organs in animals and humans (*Abotsi et al., 2011*).

Investigation of renal and hepatic functions are considered as crucial and reliable test in the toxicity study. As we know plant extracts contains various phytochemicals, it is possible that one or few among them react with liver and kidney enzymes or possibly due

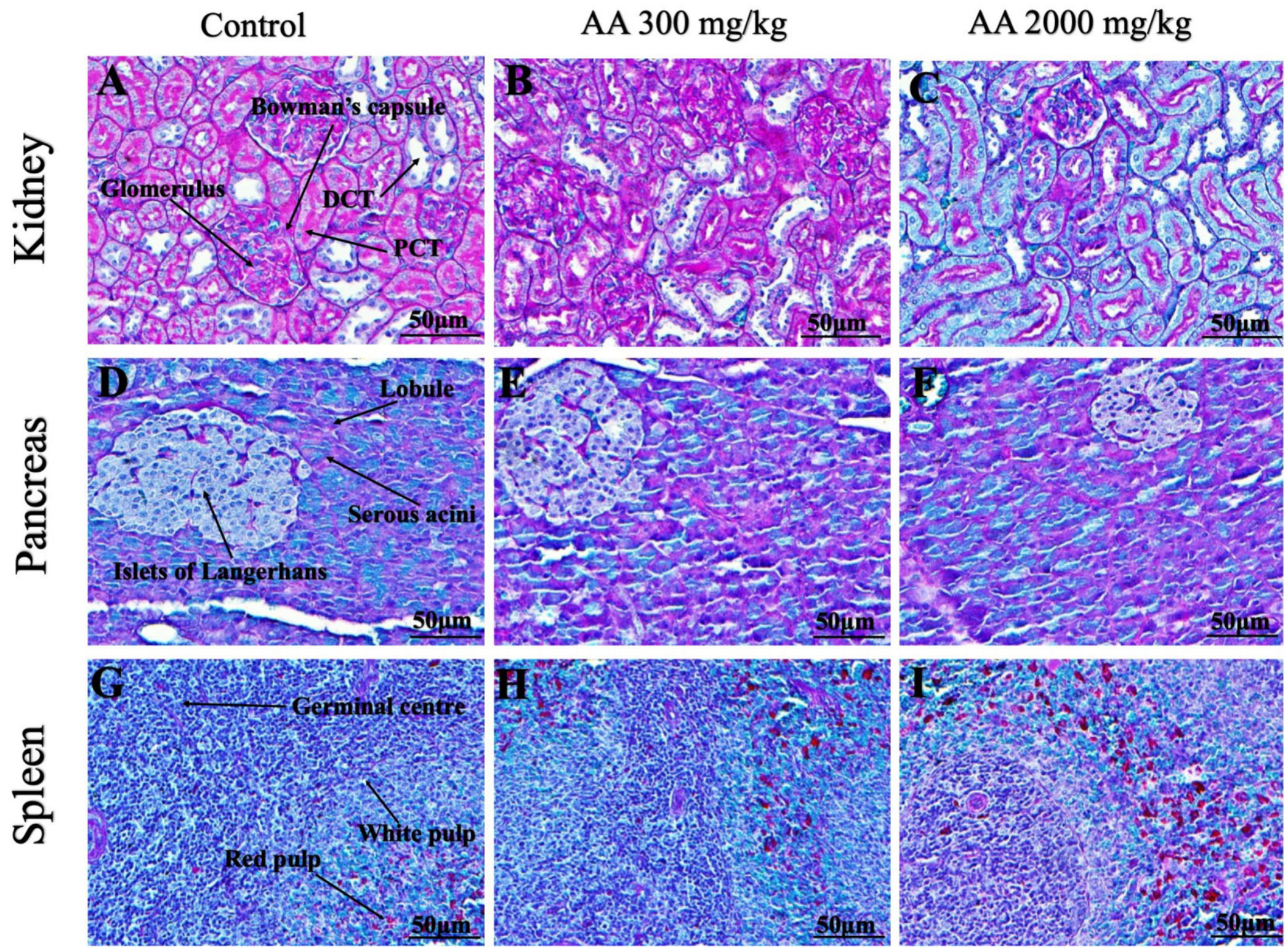

**Figure 10 Photomicrographs of vital organs demonstrating Periodic acid Schiff staining of kidneys, pancreas and spleen.** Photomicrographs of vital organs demonstrating normal architecture of kidneys (A–C), pancreas (D–F) and spleen (G–I) after a single oral dose of AA (Periodic acid Schiff stain, ×200). Proximal convoluted tubule (PCT), Distal convoluted tubule (DCT).

to the synergism (*Olorunnisola, Bradley & Afolayan, 2012*). In toxicity studies, the level of creatinine in the serum represents the clear picture to predict renal physiology. It remains to be in normal ranges, as daily synthesis and renal excretion are in equilibrium in healthy mammals (*Yilmaz et al., 2007*). In our findings, renal function tests, creatinine and urea were noted in the normal ranges which is in agreement with the previous published articles (*P'ng, Akowuah & Chin, 2013*).

Similarly, the liver enzymes, ALT, AST and AP levels were observed in the normal range after treatment with AA. These are the prominent enzymes produced by liver cells, their deregulations may lead to increase their level in the serum, thus, serve as an essential indices to elucidate inflammation, necrosis and provide clear understanding of intoxication in the liver tissues (*Imafidon & Okunrobo, 2012*; *Yakubu et al., 2003*). Moreover, AA did not alter serum albumin and globulin concentration and hepatocellular functions which depicts its safety on the liver tissue (Table 6). In animals, normal functionality of liver and kidney is important, which is in consistent with *Loha et al. (2019)* findings, that reveals no alteration to hematological and biochemical parameters in rats upon administration of *Syzygium guineense*. The outcomes on hematological and biochemical profiles suggest that AA did not cause toxicity at organ level, which act as a

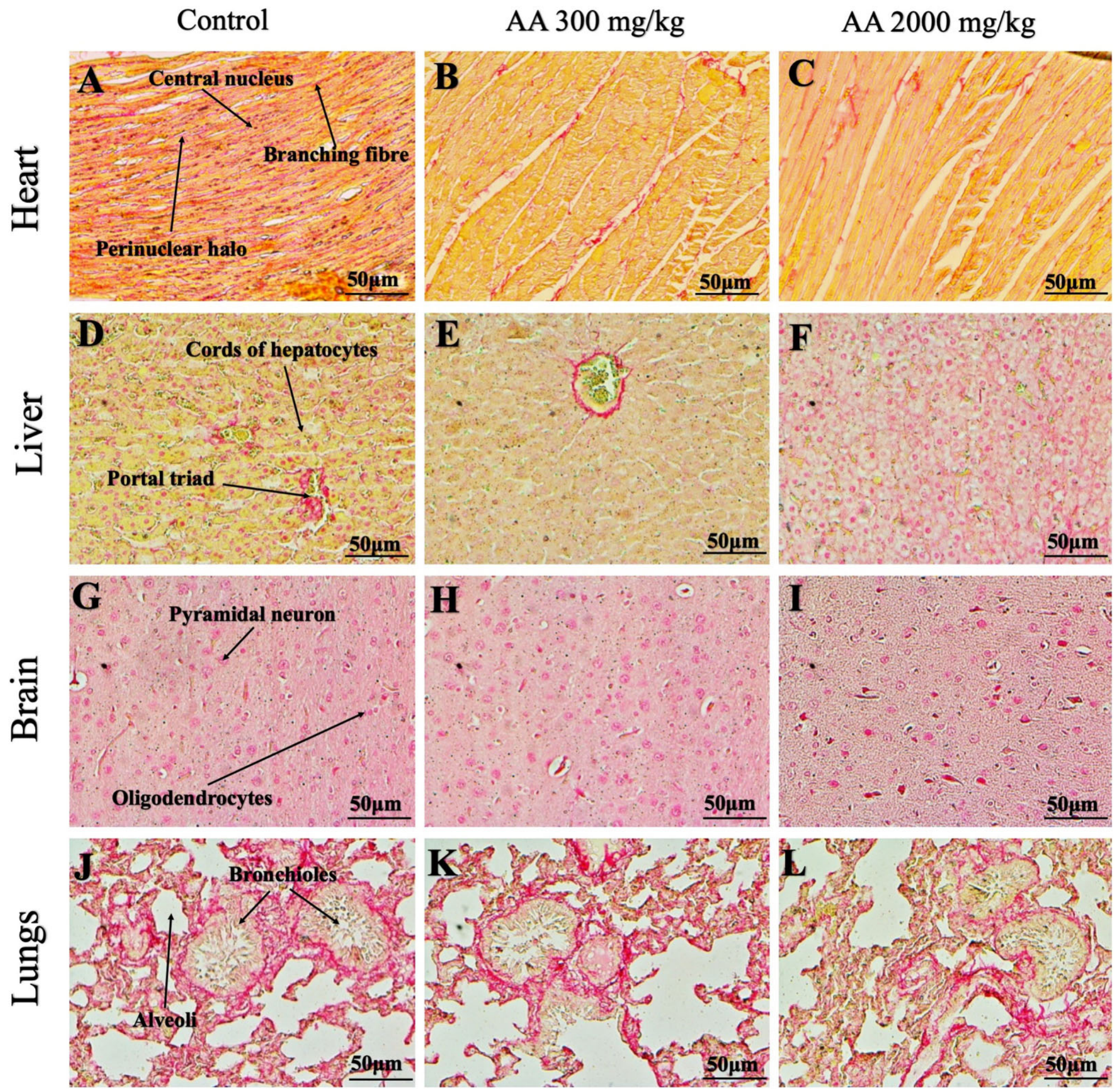

**Figure 11 Photomicrographs of vital organs representing Oil O Red staining of heart, liver, brain and lungs.** Photomicrographs of vital organs representing normal architecture of heart (A–C), liver (D–F), brain (G–I) and lungs (J–L) after a single oral dose of AA (Oil O Red stain, ×200).

basic functions test and assure serum biochemical as a crucial parameters in the toxicity study of phytochemicals (*Bariweni, Yibala & Ozolua, 2018*).

Furthermore, it is important to consider histopathological study with multiple staining techniques; H&E, MT, PAS and Oil O Red on vital organs and visualize them with appropriate microscope for in depth and critical analysis at cellular levels. In this study, the photomicrographs of heart and liver from both the treatment and control animals exhibited regular morphology and normal architecture. The heart tissues showed normal cardiac muscle fibres with centrally placed nucleus and transverse striations along with properly localized intercalated discs. Perinuclear halo, clear space around the nucleus were visible, with no signs of necrotic injury, hypertrophy, fibrosis and accumulation of fats with normal glycogen content were observed under microscope. Moreover, liver tissues displayed regular pattern in hepatocytes and polygonal shape with radiating cords, and

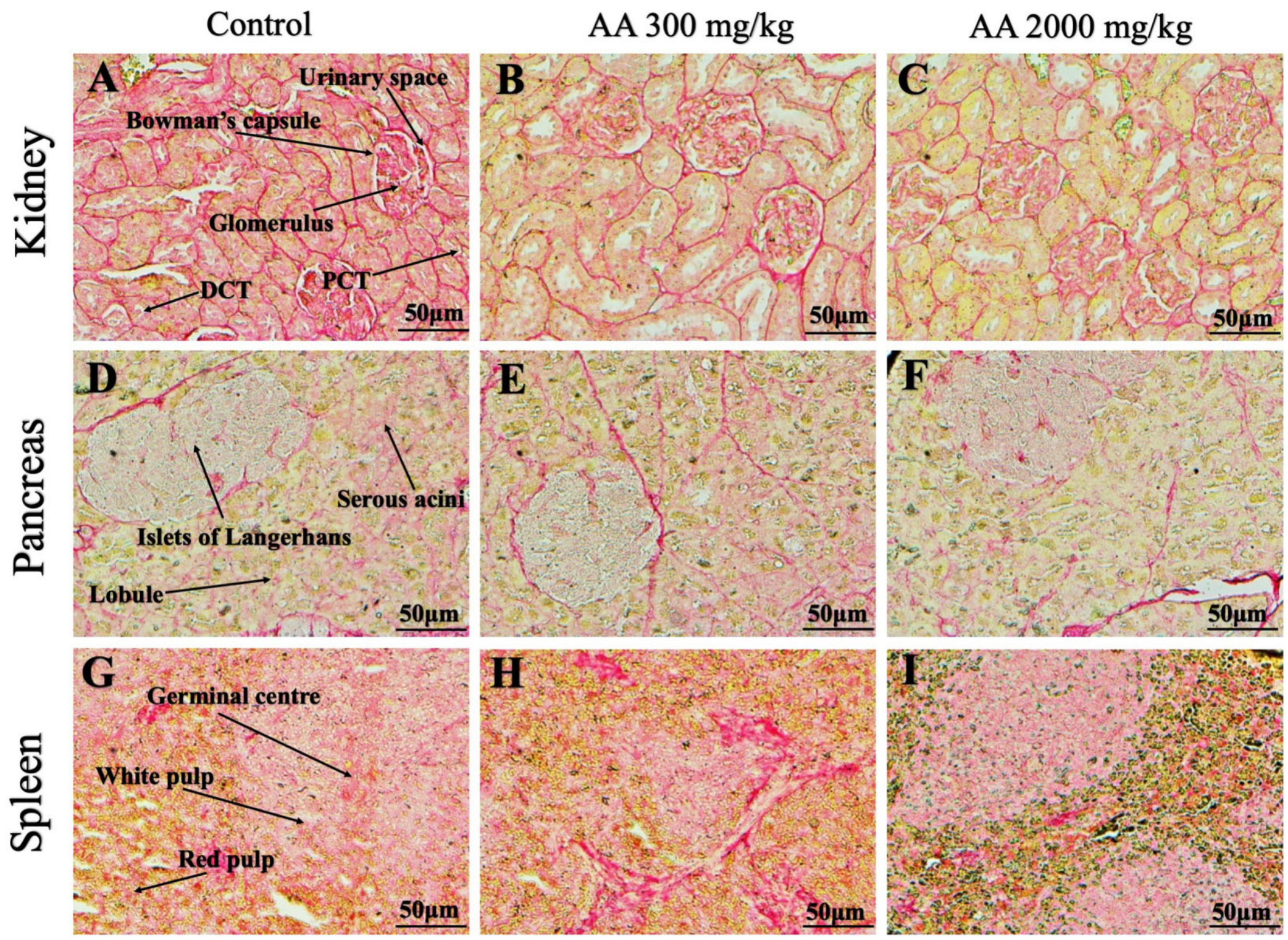

**Figure 12 Photomicrographs of vital organs representing Oil O Red staining of kidneys, pancreas and spleen.** Photomicrographs of vital organs representing normal architecture of kidneys (A–C), pancreas (D–F) and spleen (G–I) after a single oral dose of AA (Oil O Red stain, ×200). Proximal convoluted tubule (PCT), Distal convoluted tubule (DCT).

visible congestion of hepatic sinusoids. There were no lesions related with necrosis or apoptosis, no hemorrhages and fatty changes were seen around the central vein, sinusoids and portal triad. Importantly, no signs of infiltration of monocytes and macrophages were detected in the liver. Brain tissues presented the normal architecture in the different regions without any signs of morphological changes, as necrosis and distorted neurons in all the treatment and control group. The pyramidal neurons of cerebral cortex exhibited regular shape with scant and delicate cytoplasm. Glial cells were visible with a uniform outline in all the groups. Thorough microscopic analysis of lungs revealed normal honeycomb appearance of alveoli lined by flattened squamous cells, whereas, bronchioles were surrounded with epithelium and smooth muscles with no signs of congestion, necrosis and apoptotic cell death in the treatment groups.

On the other hand, kidney tissues reflected no morphological alterations in glomerular structure as well as in proximal and distal tubules. Nephrons appeared to be normal without any damage to the blood vessels and no signs of necrosis were observed in the tubules. Moreover, renal tissues displayed no accumulation of collagen, fats and glycogen in the treatment groups. In pancreas, acinar cells were normal with either round or oval shape, islets of Langerhans were visible and appeared in the form of small groups with lightly stained cytoplasm. While spleen, showed normal morphology with white and red pulp. White pulp appeared to be regular with diffused and dense distribution of

lymphocytes and red pulp showed sinusoids without any signature of apoptotic and necrotic lesions. These histopathological investigations are in line with Loha et al. (2019) and Yuet Ping et al. (2013), who reported no morphological differences in tissues of control and treatment groups.

As we know histopathological examination of vital organs and tissues serves as a basic test to judge the safety of any drug candidate (Greaves, 2011). H&E staining is most widely used to illustrates the architecture of cytoplasm and nucleus, emphasizing more toward the morphological aspects of tissues to observe any gross pathological changes (Fischer et al., 2008), whereas, MT describes the alterations in connective and soft tissue components in terms of sclerotic lesions, perivascular fibrosis and scar formation, which indicates direct damage to the vital organs due to chemical and mechanical injury (Cabibi et al., 2015). Likewise, PAS staining is another special stain to demonstrate the presence of neutral mucopolysaccharides, particularly glycogen as well as it pinpoints thickening of basement membrane in tissue structures to delineate derangements in the organization of tissues. Furthermore, Oil O Red staining reveals abnormal fat deposition in the tissues and widely employed in experimental studies to expose any metabolic abnormality due to intoxication, as the abnormal deposition of neutral fats indicates metabolic malfunctioning of the organs upon injury (Kumar, Kiernan & Dako, 2010).

Therefore, from histopathological analysis, we conclude that AA did not produce any abnormal changes and structural alterations at high dose in female SD rats. The overall outcomes of this study permit to rank AA under category 5 with low acute toxicity exposure, according to the Globally Harmonized System of Classification and Labelling of Chemicals (OECD guidelines 423) (OECD, 2001). These results are in agreement with the study conducted by Ng'uni, Klaasen & Fielding (2018), who showed that *Galenia africana* extract can be categorized under cetagory 5, when administered orally in the single dose of 2,000 mg/kg in female Sprague Dawley rats and is in compliance with Globally Harmonized System of Classification and Labelling of Chemicals. Previous findings support this study by establishing AA as a safe candidate up to 2,000 mg/kg after single dose administration.

However, in vitro testing on cell lines and other alternative methodologies are the limitations of this study, as careful and in-depth data validation is required to meet high standards of safety evaluation. It is difficult to establish a safe starting animal dose through in vitro testing. Thus, it appears crucial and of prime importance to perform *in vivo* toxicity testing in order to accomplish safety profile of the drug candidates prior to the drug development. Importantly, $LD_{50}$ obtained from animals can only provide an effective dose concentration for humans in preclinical phase, as it is practically impossible to perform acute toxicity on humans from ethical point of view. In the context of future study, we intend to perform further study on the disease model and for that we need to have complete toxicological data from rodent model with the possible therapeutic dose.

## CONCLUSIONS

Our current toxicological investigation suggest that AA is non-toxic even at high dose, it neither exhibited any alterations to the hematological and biochemical parameters nor showed any signs of physical or behavioral anomalies. Moreover, no mortality or major

histopathological changes was observed in treated rats when compared with control groups, thus establishing AA as a safe medicinal agent in appropriate dosages. However, an in-depth study on the subacute/chronic disease model to determine its potential therapeutic role on targeted ailments is warranted.

### Funding

This study is financially supported by Taylors University Flagship Research Grant (TUFR/2017/002/01) under the umbrella of Ageing and Quality of Life. The funders had no role in study design, data collection and analysis, decision to publish, or preparation of the manuscript.

### Grant Disclosures

The following grant information was disclosed by the authors:
Taylors University Flagship Research Grant: TUFR/2017/002/01.

### Competing Interests

The authors declare that they have no competing interests.

### Author Contributions

- Khurram Aamir conceived and designed the experiments, performed the experiments, prepared figures and/or tables, authored or reviewed drafts of the paper, approved the final draft.
- Hidayat Ullah Khan performed the experiments, prepared figures and/or tables, approved the final draft.
- Chowdhury Faiz Hossain authored or reviewed drafts of the paper, approved the final draft, extraction and isolation.
- Mst. Rejina Afrin authored or reviewed drafts of the paper, approved the final draft, isolation and purification of arjunolic acid.
- Imam Shaik analyzed the data, authored or reviewed drafts of the paper, approved the final draft.
- Naguib Salleh conceived and designed the experiments, authored or reviewed drafts of the paper, approved the final draft.
- Nelli Giribabu analyzed the data, authored or reviewed drafts of the paper, approved the final draft.
- Aditya Arya conceived and designed the experiments, contributed reagents/materials/analysis tools, authored or reviewed drafts of the paper, approved the final draft.

### Animal Ethics

The following information was supplied relating to ethical approvals (i.e., approving body and any reference numbers):

The Institutional Animal Care and Use Committee (IACUC), Faculty of Medicine, University of Malaya approved this research (Ethics No. 2018-210605/TAY/R/AA (2018294)).

## Data Availability

Raw data is available in a Supplemental File.

A voucher specimen of the dried bark is available at the East West University herbarium (voucher # EWUH-PHRM-180001).

## Supplemental Information

Supplemental information for this article can be found online at http://dx.doi.org/10.7717/peerj.8045#supplemental-information.

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
