# Peer review of "Oral toxicity of arjunolic acid on hematological, biochemical and histopathological investigations in female Sprague Dawley rats"

_PeerJ, doi:10.7717/peerj.8045_

## Round 0.1 · original submission · Major Revisions

Based on the advice received, I have decided that your manuscript could be reconsidered for publication should you be prepared to incorporate major revisions suggested by the reviewers.

I would ask you to respond at all to the points raised by the reviewers. Please pay particular attention to data presentation and the lack of some important details in the material and methods section.

·

Basic reporting

Overall, the document is well-organized and well-structured. The references are adequate concerning the subject of the study. However, the study has some English grammar mistakes that must be corrected prior to its acceptance. Besides, several issues concerning the quality of the images, which did not achieve an optimal resolution, were observed.
It is also important to mention that in the whole manuscript, the authors used inappropriate words to describe the results (for instance, malicious, insignia, extraordinary, and exceptional). Please, in order to keep a high standard level of technical language, replace such words for a more scientific vocabulary.

Experimental design

The exact geographic coordinates in where the plant was collected must be informed; how the authors identified the obtained crystals as the AA? Which analytical procedures were applied to confirm such data?
The lack of alternative methodologies is a critical problem of the study! Did the authors consider testing the AA using in vitro techniques? There are several possibilities that could be employed to evaluate the potential toxicity of AA (hemolysis test, HET-CAM test, culture cells). The potential lacking of such data must be at least recognized by the authors;
Details regarding the expression of biochemistry and hematological results are misleading. Besides, the parameters used to perform the histopathological observation must be informed. The specific reasons for each staining should be mentioned. Please, check these points.
In the statistical analysis section, the assigned post-test and the data normality evaluations were not described. The authors should include these information.

Validity of the findings

In the discussion section, the authors exhaustedly repeated the results instead of correlating them with the scientific data. The real contribution and novelty of the study were not highlighted; Additionally, a paragraph regarding the limitations of the study must be included (the lack of alternative methodologies); In my opinion, more information regarding the pharmacological actions of AA should be included in order to reinforce its potential applications.

Additional comments

Abstract – the dosage administered in the animals is not clear; the statement is confusing, please, check; the experimental groups are also difficult to understand.
Avoid using the word sacrifice; minor English issues that must be verified;
Introduction – This section is well-organized and the objectives are clear; Minor English issues;
In the results section, the organs selected to histopathological evaluation are mentioned, which did not occur in the material and methods. Please, correct it.
The figure 1, 4, 5 and 7 did not achieve an optimal resolution and quality. It is difficult to understand and identify the main structural features of all tissues. Moreover, for ethical purposes, the authors must avoid using images of real animals (figure 1). Please, modify it.

·

Basic reporting

The text is original, but English needs to be corrected mainly due to the lack of articles preceding a few words such as line 161 “maximum”, line 212 “abnormal”, line 239 “Gradual”.

The content of figures 4, 5, 6 and 7 support the study proposal, but the low resolution of the images, as well as the size and quality of the structural indications, make it impossible for the reader to identify the structures shown correctly. The figure quality needs to be increased, as well as increase the font of the letter of the statements.

Experimental design

The work fits the scope of the journal and the methods are well described.

Validity of the findings

Although the authors have described their findings, the lack of quality does not allow a good understanding of what happened. Therefore, the figure needs to be improved.

Additional comments

The figure quality needs to be improved, and the tables are part of supplementary material, requiring only the description and discussion of some relevant results.
Please describe what the author Naguib Salleh accomplished in the study.

---

## Round 0.2 · accepted · Accept

I have received the reviews on your manuscript, based on the advice received and my own evaluation, I am happy to inform you that your manuscript could be accepted for publication in PeerJ in its present form.

·

Basic reporting

The current version of the manuscript has superior quality; English issues were solved and the writing is clearer in comparison to the previous version of the document. The authors added essential data regarding the pharmacological applications of the phytochemical ingredient, which is the main purpose of the study. They also improved the resolution of the figures and properly identified the structures in the histological evaluation. Overall, the changes made in the document significantly contributed to improving the quality of the manuscript.

Experimental design

All the questions raised by me were properly answered. The exact geographic coordinates of the sample collection and the statistical evaluation were included.

Validity of the findings

The discussion section improved; it has a suitable scientific background, which supports the findings of the study.

Additional comments

Overall, the current version of the manuscript achieves the necessary quality level regarding the scientific content. The authors carefully attended to all my comments and improved the document. I have no additional comments.

·

Basic reporting

The text has been improved, and previously found errors have been fixed.

Experimental design

The manuscript is within the scope of the journal, and the results are unpublished.

Validity of the findings

The quality of the images has been considerably improved, which allowed better identification of the histological observations.

Additional comments

The quality of the text and images has been improved, which allows a better analysis of the findings of this study.